# Ionic Chiral Ferrocene Doped Cholesteric Liquid Crystal with Electronically Tunable Reflective Bandwidth performance

**DOI:** 10.3390/ma15248749

**Published:** 2022-12-08

**Authors:** Wan-Li He, Ya-Qian Zhang, Wen-Tuo Hu, Hui-Min Zhou, Zhou Yang, Hui Cao, Dong Wang

**Affiliations:** School of Materials Science and Engineering, University of Science and Technology Beijing, Beijing 100083, China

**Keywords:** cholesteric liquid crystal, wide wavelength reflection, ferrocene, electrical tuning

## Abstract

Cholesteric liquid crystals (CLC) were widely used in optical devices as one-dimensional photonic crystals. However, their reflective bands cannot be adjusted, which limits their wide application in many fields. In this paper, a series of ionic chiral ferrocene derivatives (CD-Fc^+^) as dopants were designed and prepared, and their doping into negative liquid crystal matrix was investigated to develop cholesteric response liquid crystal composites with electrically tunable reflective bands. The effects of electric field frequency, voltage, retention time of voltage and molecular structure on the broadening of reflection bandwidth were investigated in detail.

## 1. Introduction

Cholesteric liquid crystal (CLCs) is a kind of liquid crystal mesophase with a self-organized periodic spiral structure. Since it satisfies the Bragg reflection condition, its periodic spiral structure can selectively reflect the incident light. Cholesteric LC materials, especially CLCs with wide wavelength reflection characteristics, have shown good application prospects in laser protection, intelligent switchable reflection window, light brightening film, electronic paper, infrared shielding and other fields because of their unique circular dichroism [1,2,3,4].

As we all know, the reflection wavelength of CLCs satisfies the formula: $\lambda_{P}=\;\bar{n}\cdot P_{0}$, where $\bar{n}$ is the average birefringence of LC molecules and P_0_ is the pitch of CLCs. The reflected bandwidth Δλ is related to birefringence Δn (Δn = n_e_ − n_o_) and Δp: Δλ = Δn*Δp. The Δn in common CLCs is generally within a value of 0.4, so the bandwidth of the selective reflection spectrum of a single pitch LC material is often limited to 100 nm. Since the adjustable range of Δn is very limited, researchers thus often change Δp to adjust the reflection bandwidth Δλ. Based on the principle of constructing multiple helical structures with different pitch lengths with random or gradient distributions in CLC, researchers can obtain CLCs with wide wavelength reflection in different LC systems. This can be done in several ways, including as multi-layer stacking systems, doping responsive chiral molecules, controlling the rate of photo-polymerization, using thermally induced molecular diffusion, constructing two chiral mesophases coexistence systems, memory anchoring effect of templates and molecular diffusion induced by electric or magnetic fields.

In our previous report, we used light induced molecular diffusion to construct polymer cholesteric LC networks as LC templates. We then refilled LC materials to obtain CLCs films with a reflection bandwidth of more than 1000 nm [5]. We synthesized chiral binaphthol derivatives with large temperature sensitivity, and then polymerized them at two temperature points with a large difference between high temperature and low temperature. This realized the adjustment of the pitch of CLCs and broadened their reflection bands [6]. We used photo responsive azo derived chiral compounds in chiral nematic LCs which were gradually polymerized under ultraviolet light. A polymer LC film capable of reflecting a wavelength range of 1000–2400 nm was obtained [7]. This has become a popular way to broaden the reflective wavelength width of CLCs, either to induce the change of the helical torsional force of chiral compounds or to form concentration gradient by external stimulation. At the same time, the polymer network was used to fix the non-uniform distribution of pitch. However, due to their permanently fixed spiral structure, the reflection bandwidth cannot be adjusted, which limits the applications in many fields.

The dynamic tuning of CLCs in reflective bands has become a hot topic because of its wider use. In general, common stimulus signals such as heat [8], light [9,10,11] and electric field [12,13,14,15,16] can be used to adjust the central selective wavelength or reflection bandwidth of CLCs. However, relatively speaking, the photo-thermal signal and humidity in the tuning process of LC devices are often accompanied by slow response speeds, low tuning accuracy and other shortcomings. They are also vulnerable to interference from the external environment. Therefore, compared with other methods, electrically tuned LC devices have the advantages of high precision and rapid response, and the equipment is easy to integrate into electronic equipment.

In this paper, we designed a series of ionic chiral ferrocene derivatives (Figure 1, CD-Fc^+^) as liquid crystal dopants and investigated the performance of electronically tunable reflection bandwidth after doping into the negative liquid crystal matrix in order to develop practical CLC composites with electrically tunable reflective bands. The effects of DC voltage intensity, DC voltage holding time and molecular structure on the broadening of reflection bandwidth were studied in detail.

## 2. Materials and Methods

The oxidant, tetrafluoroborate nitrous oxide (NOBF_4_), (R)-(+)-Binaphthol ([a]_D_^25^ = +35.2° (c = 1.0 in THF)), Ferrocene formic acid or dicarboxylic acid and other chemical reagents in this work were commercially available from Energy Chemical Co., Ltd. (Shanghai, China). or from Beijing Chemical Product Factory (Beijing, China). They were used as supplied, without further purification. Nematic LC matrix (BHR28000-300, Δn = 0.149, Δ = −9.6, T_N-I_  =  81 °C) was obtained from Beijing Bayi Space LCD Materials Co., Ltd. (Beijing, China). The right-handed chiral dopant R5011 (HTP ~110 μm^−1^) was purchased from Jiangsu Hecheng Display Technology Co., Ltd. (Nanjing, China). The CD-Fc molecules consisted of a chiral binaphthol unit and a ferrocene unit and were prepared by a simple esterification reaction as shown in Figure 1. The detailed synthesis steps are described in the Appendix A.

The structure information of CD-Fc molecules were characterized by ^1^H NMR spectra with a Bruker AV400 instrument. Deuterated chloroform was the solvent and tetramethylsilane (TMS) was the internal standard. Matrix-assisted laser desorption/ionization time-of-flight (MALDI-TOF) mass spectrometer was operated on a Bruker solanX 70 FT-MS. Infrared spectra were recorded by using a Nicolet 510P FT-IR spectrometer. X-ray photoelectron spectroscopy (XPS) was implemented with a Thermo Scientific K-Alpha scan with Al Kα rays (6 eV) as the excitation source. The Fe valence states of the CD-Fc^+^BF4 powder samples before and after oxidation were characterized by XPS within the Fe 2p region. The arbitrary wave generator (UTG932) and voltage signal amplifier (PINTECH HA-400) were used to provide the external electric field conditions required in the experiment, and the electronic tuning of the reflective bandwidth of the LC system was also realized on the APS3005D potentiostate. The texture images of the LC system under different applied voltages were observed with a polarizing microscope (Olympus BX51). The motion laws of negative LC molecules and CD-FC^+^ chiral compounds under the applied electric field were analyzed. The reflective bandwidth, central selective reflective wavelength and transmissivity of the LC system were measured by ultraviolet visible spectrometer (JASCOO-V570).

Two pieces of ITO glass (size: 1.0 cm × 1.5 cm, square resistance: ~20 Ω) were cleaned as the substrate. Then the conductive surfaces of the two substrates were placed face to face and offset by 5 mm from each other. Then, 40 μm polyethylene film spacers were sandwiched between the two substrates near the edges of the two non-offset sides. The sides were glued with the two substrates to form a LC cell. Finally, two conductive leads were pasted on the conductive surface of the offset side (5 mm area) of the two substrates with conductive tape. According to the design proportion in Table 1, the LC matrix, chiral R5011 and ferrocene derivative CD-Fc were added into 10 mL clean glass bottles, respectively, and then fully dissolved with a small amount of dichloromethane. Then, 2.0 g/L ethanol solution of NOBF_4_ was prepared, and the NOBF_4_ solution with the same molar amount as the above-mentioned CD-Fc was weighed out and added into the bottle. The glass bottle was vortex-vibrated for 1 min and ultrasonic-vibrated for 10 min until the mixture in the bottle was fully mixed. Then the glass bottle was placed in a vacuum drying oven at 60 °C and dried for about 10 h. After the solvent in the mixture was completely removed, the mixture was poured into the LC cell through capillary action.

The observation of electronic tuning reflection bandwidth of all LC samples was carried out according to the following steps, as shown in Figure 2. Firstly, in the initial state, the CD-Fc^+^ in the cholesteric liquid crystal without electric field was uniformly distributed. The pitch of the cholesteric phase was uniform and single and the LC molecules were periodically arranged, showing a standard planar texture and reflecting narrow circularly polarized light. Then, when the DC electric field in the direction of the parallel spiral axis was applied to the LC cell for a certain time, the cationic CD-Fc^+^ diffused and migrated to the negative electrode of the ITO glass substrate under the electric field. The content of the chiral compound CD-Fc^+^ would then have a certain concentration gradient. That is, it gradually decreased from top to bottom. At the same time, the migration of anion BF4^−^ was opposite to that of CD-Fc^+^. Since the anion did not possess a chiral structure, its migration in the electric field cannot affect the pitch distribution. The negative LC molecule BHR28000-300 showed a dynamic scattering state due to the disturbance of anion and cation migration. The LC texture plane texture was broken into disorder, showing a focal cone texture. This state can be maintained for a long time after the external electric field was turned off. Then, after fast application of high-frequency AC electric field (τ_AC_ = 2 s, f = 100 kHz), the LC molecules would be parallel oriented again, caused by the negative dielectric anisotropy of the LC. However, CD-Fc^+^ with a certain concentration gradient cannot be recovered immediately. This resulted in a pitch gradient in the plane state, and a wide reflection band of the sample. This can reflect a certain range of circularly polarized light. On the other hand, when an opposite DC electric field was applied to the upper LC cell for a certain time, CD-FC^+^ would be uniformly dispersed into the LC phase again. Due to the migration of anions and cations, the planar texture of LC molecules was destroyed again, showing a light scattering focal cone state. After fast application of high-frequency AC electric field again (τ_AC_ = 2 s, f = 100 kHz), the LC molecules tended to be arranged parallel to the substrates again. They would return to the initial single pitch arrangement and the reflection band would also return to the initial state. After high frequency AC electric field was applied to the LCD cell again quickly (τ_AC_ = 2 s, f = 100 kHz), the LC molecules tended to be arranged parallel to the substrate again, showing a single pitch arrangement, and their reflective band returned to the initial state. In summary, by applying electric fields of different frequencies or directions, the LC mixture can switch between three different optical states: transparency, scattering and broadband reflection. After each electric field was turned off, each corresponding state can remember the state for a certain time.

## 3. Results

The process of electric field tuning reflection wavelength of cholesteric LC sample (such as sample 5) is shown in Figure 3. Under the action of DC electric field (I→II, E_DC_ = 40 V), the chiral compound CD-Fc^+^ in the cholesteric LC diffuses and migrates to the ITO glass substrate connected to the negative pole of the power supply, forming a gradient distribution. The LC molecules were disorderly arranged under the disturbance of surrounding ions, and the LC texture presented a light scattering focal point cone state. High frequency AC electric field (Ⅱ→Ⅲ, E_AC_ = 40 V@100 KHz) was then applied. Negative LC molecule BHR28000-300 had a parallel orientation, CD-Fc^+^ retained gradient distribution in the plane state and the reflection wavelength band was effectively widened from 250 nm to 630 nm. Then, the reverse DC electric field (E_DC_ = −40 V) was applied to the LC system (Ⅲ→Ⅳ), CD-Fc^+^ was uniformly dispersed into the LC matrix, and the high frequency AC electric field was used to promote the parallel orientation of LC molecules. CLC returned to the original single pitch plane state and the reflective wavelength width was close to the original state (~250 nm). The difference was that the central reflection wavelength was 1850 nm in the initial state, and after tuning, it shifted to 1910 nm. It was not completely restored to the initial state. This may be due to the fact that a small amount of CD-Fc^+^ was adsorbed on the ITO glass substrate, that is, a small amount of CD-Fc^+^ was separated from the LC mixture, resulting in a decrease in the content of chiral compounds.

It was found that the DC electric field (E_DC_ = 40 V, τ_DC_ = 25 s), applied high-frequency AC electric field (E_AC_ = 40 V@ f = 100 kHz) and the range of reflection wavelengths tuned by the electric field for each sample was different, as shown in Figure 4b. Among all chiral ions, the CD-Fc^+^2-BenC7 and CD-Fc^+^-BipOC8, with long alkyl chains, had better solubility in the LC matrix. In particular, the ferrocene structure in CD-Fc^+^2-BenC7 contained a longer alkyl chain, which can further improve its solubility in LC matrix. This was consistent with the observation result of each sample in POM. It can be seen from the figure that there were small amounts of insoluble substances in the plane texture of sample 1 and sample 3, while only sample 2 showed good compatibility and the plane texture was relatively complete. Insoluble chiral ions could not be effectively electro-migrated and chiral-induced in the LC matrix, so the broadening of the reflection wavelength range was limited.

As we know, when a high-frequency AC electric field was applied to negative LC (Δε < 0), the director of LC molecules tended to be perpendicular to the electric field. It was found that increasing the intensity of high-frequency alternating electric field (f = 100 kHz) was conducive to the regular arrangement of LC molecules in the direction of vertical electric field. That is, the intensity of alternating electric field had a certain positive effect on the arrangement of LC molecules parallel to the substrate. It can be seen from Figure 5 that with the increase of the AC voltage, the transmittance of sample 1 increased continuously, and the focal cone texture of light scattering gradually tended to the complete plane texture. However, it was worth noting that the intensity of the AC electric field had no obvious effect on the CLC reflection band. When AC electric fields of different intensities to sample 1 were applied with a CD-Fc^+^ concentration gradient, the chiral ion CD-Fc^+^ was not driven to redistribute or homogenize. The central reflection wavelength of sample 1 was always about 2000 nm and the reflection wavelength width was 570 nm.

The influence of DC voltage with different intensities on the reflection wavelength range of the sample was observed, as shown in Figure 6. The initial reflected wavelength width of sample 1 was 290 nm. DC voltage (τ_DC_ = 25 s) increased from 0 to 40 V, and the driving force of the electric field on CD-Fc^+^–BenC7 gradually increased, promoting the diffusion of chiral compounds in the LC system, and the pitch gradient in CLCs increased accordingly. When the voltage was increased to 40 V, the reflection bandwidth reached 570 nm. In addition, the same trend had also been found in sample 5, which contained several chiral ions CD-Fc^+^ with different structures. With an increase of DC voltage (τ_DC_ = 25 s) from 0 to 40 V, the reflection wavelength range of sample 5 was widened from the initial 250 nm to 630 nm, showing a relatively significant broadening effect. This may be due to different migration velocities in the electric field and different HTP values of chiral ions with different structures. When the DC voltage was low, the migration range of all CD-Fc^+^ was small, while when the voltage was high, more chiral ions would migrate under the driving force of the electric field. However, due to different migration speeds, chiral ions with different structures diffused to form different chiral concentrations in the diffusion direction and thus induced different pitch gradients. Due to different HTP, chiral ions with different structures diffused to form different chiral helices in the diffusion direction. They also induced different pitch gradients. As a result, the synergistic effect of migration velocity and HTP could form a larger pitch gradient, resulting in a wider reflective wavelength range.

In addition, it was found that the holding time of applying a DC electric field (τ_DC_) to the LC sample would also affect the final reflection wavelength range. As shown in Figure 7, when the DC voltage applied to sample 1 was gradually increased from 0 s to 25 s, its reflection wavelength width became wider with the increase of voltage holding time. Similarly, when the DC voltage applied to sample 5 was increased to 33 s, the reflective wavelength range obviously increased. In this process, when the voltage of the DC electric field was very small, the driving force of the chiral ions was very limited. As a result, only a few ions were driven by the electric field. Even if the holding time of the low voltage was prolonged, the migration range of the chiral ions was still very small, so the wavelength range of the reflection was only widened a little. After the driving voltage was increased, an increasing number of chiral ions can be driven by the electric field to diffuse to the cathode side. With an increase of holding time, the gradient distribution of chiral compounds became more and more obvious, thus forming a corresponding large pitch gradient. When the voltage holding time of the DC electric field was further increased, due to the small HTP values of the chiral ions, the change of the pitch gradient formed by diffusion was small, so the reflection wavelength range was increased slightly. When the applied DC voltage was kept for a certain time, the driving force of the electric field on the chiral ions and the diffusion force in the direction of the counter electric field gradually reached a balance. Then, the chiral concentration gradient did not obviously change, resulting in the lack of change in the range of the reflective wavelength of the cholesteric LC.

## 4. Conclusions

In this study, we prepared a series of ionic chiral ferrocene derivatives (CD-Fc^+^) and used them as liquid crystal dopants to develop chiral ion/liquid crystal materials with electronically tunable reflective bandwidth performance. The effects of DC voltage intensity, DC voltage holding time and molecular structure on the broadening of reflection bandwidth of host liquid crystals were systematically investigated. It was found that after doping ion type chiral compound CD-Fc^+^, its reflection wavelength range can be broadened by electric field tuning. When doping the mixtures of chiral ions with different structures, the reflection wavelength range can be broadened from 250 nm to 630 nm by electric field tuning. Then, when the electric field was applied in the opposite direction, the reflective wavelength range of the LC system can be reversibly restored to the initial state. It was also found that a higher DC electric field was beneficial to promote the gradient distribution of CD-Fc^+^ in the LC matrix. When the DC voltage was increased to 40 V, the cholesteric LC had a wider reflection wavelength range. Prolonging the holding time of the DC electric field was beneficial to promoting the migration of CD-Fc^+^ to form a concentration gradient, and the reflective wavelength range would be significantly widened. However, the solubility of ferrocene derivatives and chiral ferrocene ions in liquid crystals is low, so it is necessary to optimize the molecular structure of ferrocene derivatives in the near future to improve the solubility. Nevertheless, ionic chiral ferrocene liquid crystal systems exhibit good performance in the range of electrically tunable reflective wavelengths, which is helpful to further develop electrically tunable cholesteric liquid crystals for practical optical device applications and expand the application field of wide wavelength reflective liquid crystal materials.

## Figures and Tables

**Figure 1 materials-15-08749-f001:**
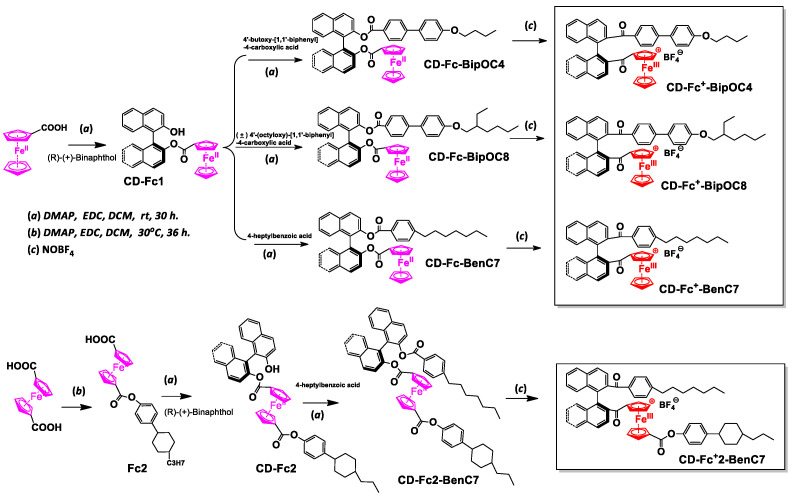
Synthetic routes of ferrocene chiral derivatives and ferrocene chiral ions (a) DMAP, EDC, rt, 30 h; (b) DMAP, EDC, 30 °C, 36 h; (c) NOBF_4_.

**Figure 2 materials-15-08749-f002:**
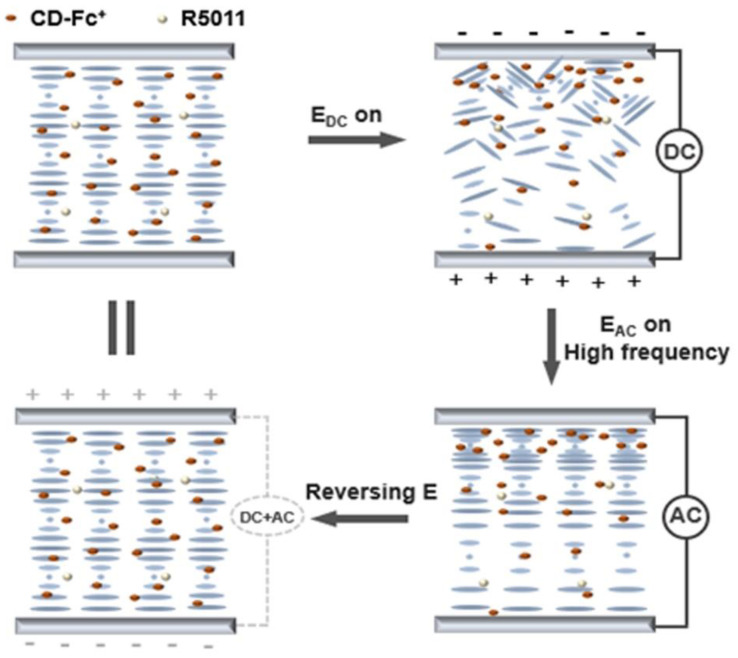
Schematic of reflection bandwidth of electronically controlled LC.

**Figure 3 materials-15-08749-f003:**
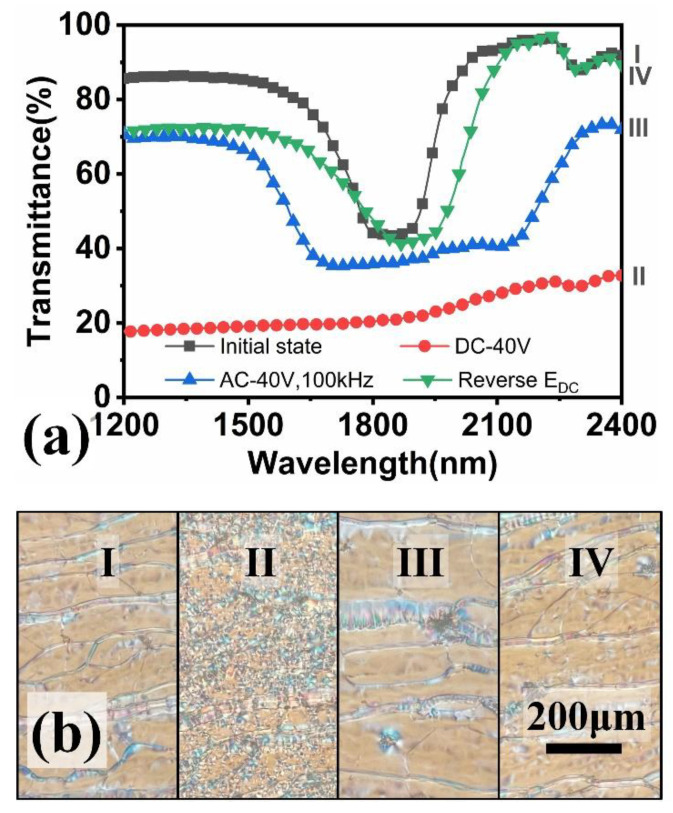
Ultraviolet transmission spectrum (**a**) and corresponding POM texture picture (**b**) of sample 5 during electric tuning (I initial state, II with DC electric field (E_DC_ = 40 V), III with high-frequency AC electric field (E_AC_ = 40 V@ f = 100 kHz), IV with DC electric field (E_DC_ = −40 V)).

**Figure 4 materials-15-08749-f004:**
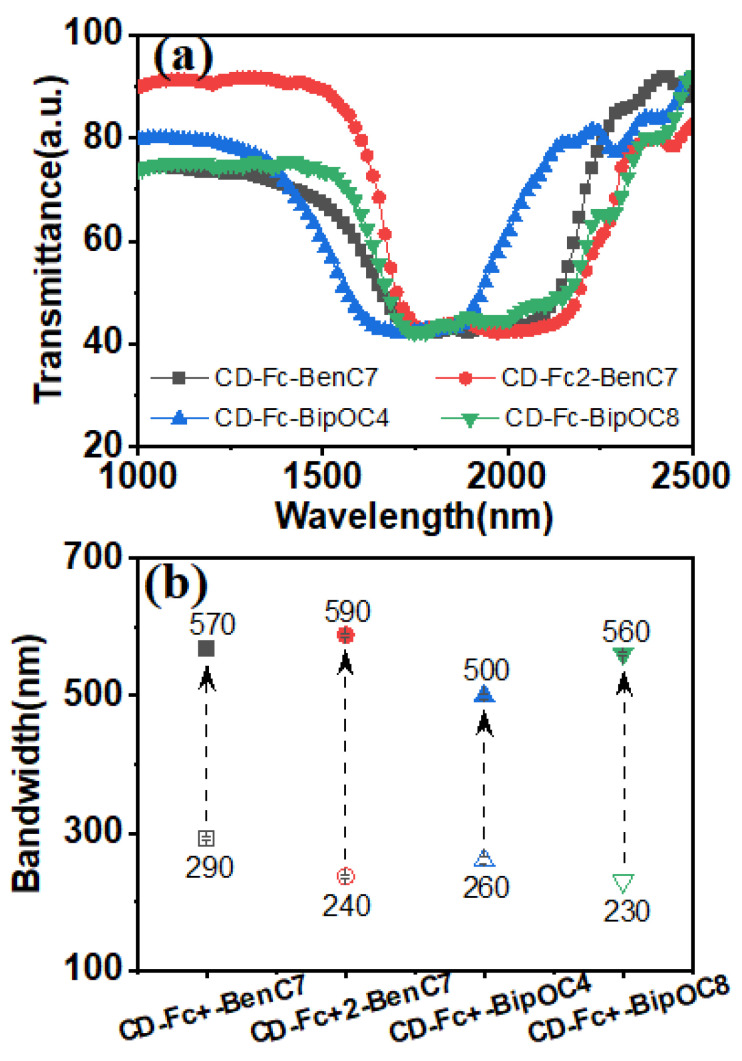
(**a**) UV transmission spectra (**b**) electronically controlled reflective bandwidth of LC samples doped with CD-Fc^+^ of different structures.

**Figure 5 materials-15-08749-f005:**
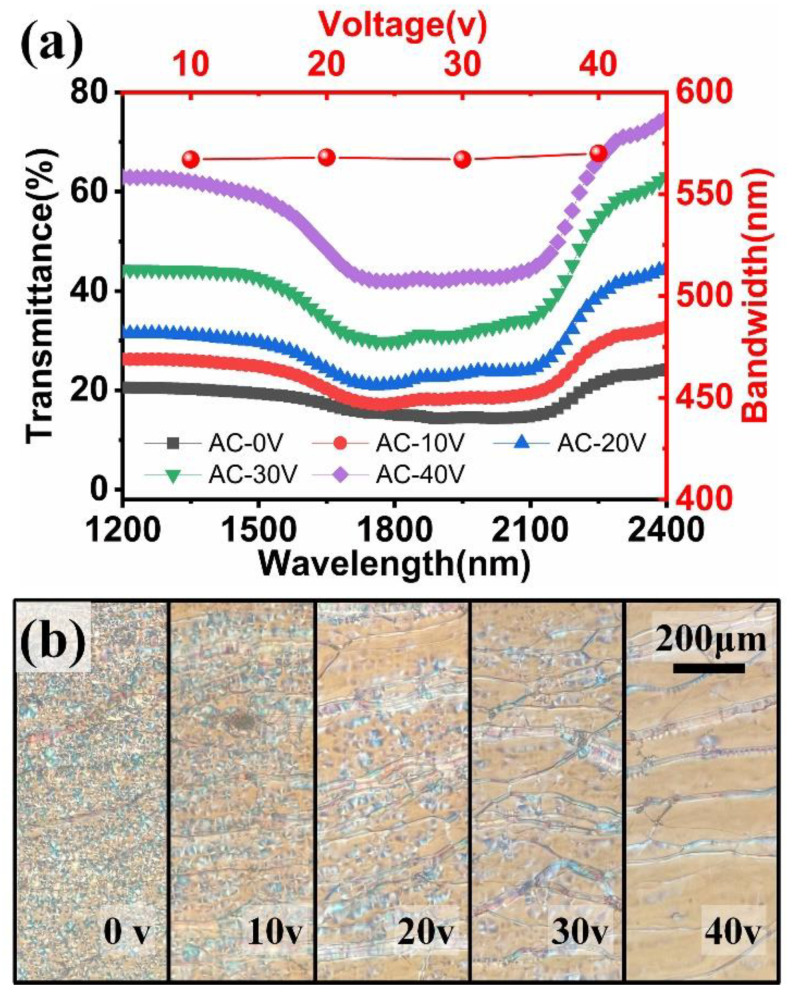
Influence of AC voltage intensity on (**a**) UV transmission spectra and (**b**) LC texture of focal cone state in LC sample 1.

**Figure 6 materials-15-08749-f006:**
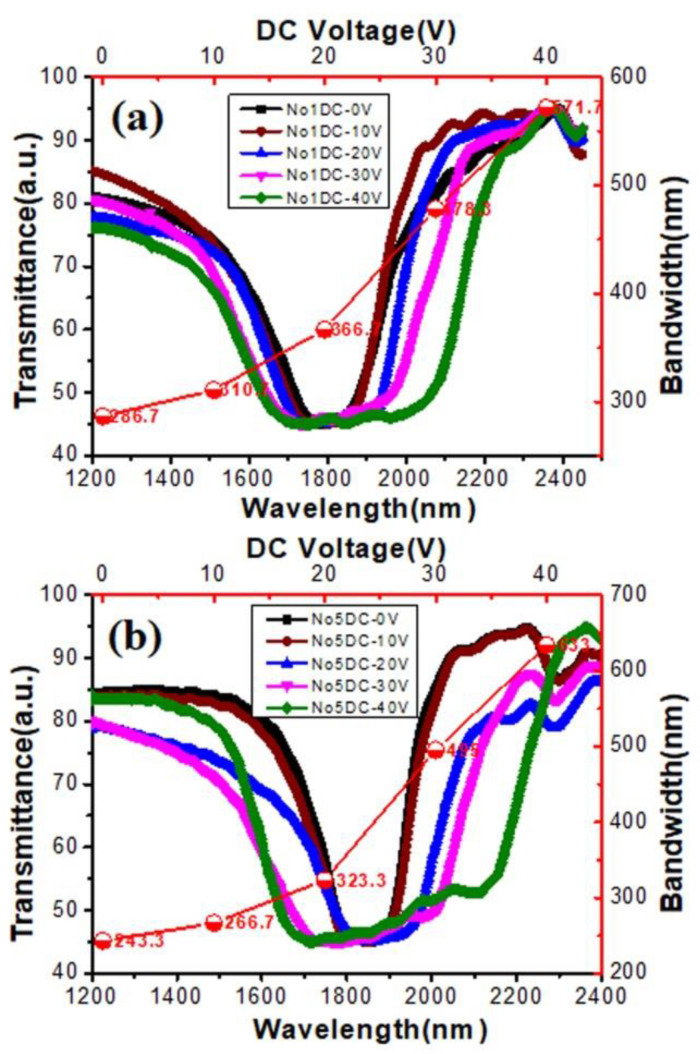
Influence of DC voltage intensity on the UV transmission spectrogram and reflection bandwidth of (**a**) sample 1 and (**b**) sample 5.

**Figure 7 materials-15-08749-f007:**
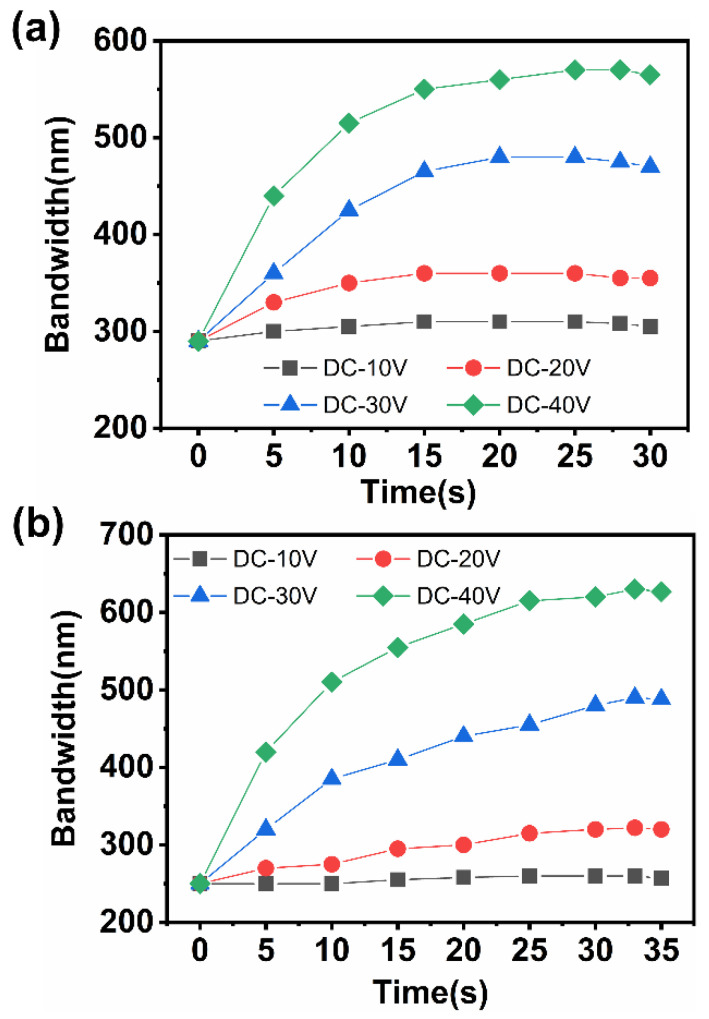
Reflection bandwidth on voltage holding time: (**a**) sample 1 and (**b**) sample 5.

**Table 1 materials-15-08749-t001:** Composition ratio of liquid crystal samples.

No.	^1^ Composition of Mixture (wt%)
1	94.0/5.0/0/0/0/0.3/0.7
2	94.1/0/5.0/0/0/0.2/0.7
3	94.15/0/0/5.0/0/0.15/0.7
4	94.28/0/0/0/5.0/0.02/0.7
5	93.2/1.0/2.0/1.0/2.0/0/0.80

^1^ Composition of mixture: BHR28000-300/CD-Fc^+^–BenC7/CD-Fc^+^2-BenC7/CD-Fc^+^–BipOC4/CD-Fc^+^–BipOC8/R5011/NOBF4.

## Data Availability

Not applicable.

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
