# Peer review of "Ionic Chiral Ferrocene Doped Cholesteric Liquid Crystal with Electronically Tunable Reflective Bandwidth performance"

_materials, 2022, doi:10.3390/ma15248749_

Round 1
Reviewer 1 Report
This manuscript reports on the synthesis of a series of chiral ferrocenium derivatives and their use as dopants to prepare mesomorphic systems with an electrically tunable reflective bandwidth. The effects of electric field frequency, voltage and molecular structure on the broadening of reflection bandwidth have also been studied.
The compounds described have been chemically characterized by mass spectrometry and 1H NMR, infrared and X-ray photoelectron spectroscopies. The results are of interest and the manuscript is well presented and organized.
The manuscript can be accepted for publication in Materials. However, before publication, the authors should consider the following points:
1. The purity of the compounds is essential in mesomorphic systems. Therefore, to ensure the purity of the prepared compounds, the authors should provide the elemental analysis of the ferrocene derivatives.
2. In figure 1, the structures of the ferrocene derivatives do not show their chiral nature. Thus, these structures should be modified accordingly.
3. Compounds CD-FC-BipOC8 and CD-FC+-BipOC8 bear a alkoxy chain with a tertiary carbon. Is this chain chiral or not? To avoid any confusion, this point should be clarified.
4. For a proper characterization, specific rotation of the chiral compounds should be provided.
5. Lines 68-70. The authors state “…we investigated the performance of reversibly dynamically tuned reflective bands of ionic ferrocene chiral compounds (Figure 1, CD-Fc+) doped with negative LCs…”. However, in lines 258-259 it states “..we prepared a class of ferrocene derived ionic chiral compound CD-Fc+, and used it as a dopant to prepare…” This apparent contradiction needs to be revised.
6. The size and resolution of POM images displayed in figures 3 and 5 should be improved.
Author Response
This manuscript reports on the synthesis of a series of chiral ferrocenium derivatives and their use as dopants to prepare mesomorphic systems with an electrically tunable reflective bandwidth. The effects of electric field frequency, voltage and molecular structure on the broadening of reflection bandwidth have also been studied.
The compounds described have been chemically characterized by mass spectrometry and 1H NMR, infrared and X-ray photoelectron spectroscopies. The results are of interest and the manuscript is well presented and organized.
The manuscript can be accepted for publication in Materials. However, before publication, the authors should consider the following points:
- The purity of the compounds is essential in mesomorphic systems. Therefore, to ensure the purity of the prepared compounds, the authors should provide the elemental analysis of the ferrocene derivatives.
Response: Thank you very much for reviewing the manuscript. The element analysis of the target ferrocene derivatives in paper, have been added to the electronic supplementary information of the manuscript.
- In figure 1, the structures of the ferrocene derivatives do not show their chiral nature. Thus, these structures should be modified accordingly.
Response: Thank you very much for reviewing the manuscript. The mentioned chiral structure of ferrocene derivatives in Figure 1 has been modified accordingly in the revised version.
- Compounds CD-FC-BipOC8 and CD-FC+-BipOC8 bear a alkoxy chain with a tertiary carbon. Is this chain chiral or not? To avoid any confusion, this point should be clarified.
Response: Thank you very much for reviewing the manuscript. The mentioned CD-FC BipOC8, CD-FC+-BipOC8 and other ferrocene derivative as wellas therir ions, have been emphasized that the benzoic acid raw materials used are achiral compounds in the synthesis route and the ESI.
- For a proper characterization, specific rotation of the chiral compounds should be provided.
Response: Thank you very much for reviewing the manuscript. The optical rotation of the mentioned chiral compound binaphthol has been added accordingly to the synthesis route and support materials of the revised manuscript.
- Lines 68-70. The authors state “…we investigated the performance of reversibly dynamically tuned reflective bands of ionic ferrocene chiral compounds (Figure 1, CD-Fc+) doped with negative LCs…”. However, in lines 258-259 it states “..we prepared a class of ferrocene derived ionic chiral compound CD-Fc+, and used it as a dopant to prepare…” This apparent contradiction needs to be revised.
Response: Thank you very much for reviewing the manuscript. The mentioned contradictions in the manuscript have been corrected in the revision of the manuscript.
- The size and resolution of POM images displayed in figures 3 and 5 should be improved.
Response: Thank you very much for reviewing the manuscript. The size and resolution of POM images both in Figures 3 and 5 have been improved in the revised version.
Reviewer 2 Report
In this work, the authors prepare a class of ferrocene chiral ions used as a dopant in a chiral ionic ferrocene derivatives/liquid crystal system allowing to electronically tune its reflection bandwidth. They study several effects on the broadening of the reflection bandwidth including electric field frequency, voltage, voltage holding time as well as the molecular structure of ferrocene derivatives. The results are interesting and will surely impact the liquid crystal community. I have some additional questions I would like to see addressed which will clarify the system behavior, but overall, I consider it suitable for publication.
- The manuscript would benefit from a thorough proofread. For example, in line 13 some words are missing and line 34 is incomplete.
- Did the authors perform any voltage cycles? It would be interesting to add this result in the Supporting Information (SI).
- In the last paragraph of the manuscript, the authors mentioned that when the voltage was held for a long time, the bandwidth was maintained. Fig 7. does not reflect this result. The authors should add their measurements after 35 sec in the SI.

Author Response
In this work, the authors prepare a class of ferrocene chiral ions used as a dopant in a chiral ionic ferrocene derivatives/liquid crystal system allowing to electronically tune its reflection bandwidth. They study several effects on the broadening of the reflection bandwidth including electric field frequency, voltage, voltage holding time as well as the molecular structure of ferrocene derivatives. The results are interesting and will surely impact the liquid crystal community. I have some additional questions I would like to see addressed which will clarify the system behavior, but overall, I consider it suitable for publication.
- The manuscript would benefit from a thorough proofread. For example, in line 13 some words are missing and line 34 is incomplete.
Response: Thank you for reviewing the manuscript. The typing errors mentioned have been corrected in the revised version.
- Did the authors perform any voltage cycles? It would be interesting to add this result in the Supporting Information (SI).
Response: Thank you for reviewing the manuscript. Due to the severe epidemic situation, we cannot conduct long-term voltage cycling stability test on the samples. It is certain that after DC and AC electric fields are applied to the liquid crystal sample, and after the reverse electric field is applied again, the sample can return to its initial state, so the sample can repeatedly respond to the electric field to adjust the reflected wavelength range.
- In the last paragraph of the manuscript, the authors mentioned that when the voltage was held for a long time, the bandwidth was maintained. Fig 7. does not reflect this result. The authors should add their measurements after 35 sec in the SI.
Response: Thank you for reviewing the manuscript. The description in Figure 7 is not accurate enough, and we have corrected it in the revised version. Due to the serious epidemic situation, we can not retest the stability of the liquid crystal samples in a long-term voltage cycle, but we can be sure that after the voltage is kept for a certain time, the driving force of the electric field on the chiral ions in the samples and the diffusion force in the direction of the counter electric field gradually balance, and the chiral concentration gradient changes no longer obviously, resulting in almost no change in the reflected wavelength range.
Round 2
Reviewer 1 Report
I would like to thank the authors for addressing my initial comments. However, the point related to the optical rotation has not been reviewed. Although the authors state in their response that "The optical rotation of the mentioned chiral compound binaphthol has been added accordingly to the synthesis route and support materials of the revised manuscript", the optical rotation values are still missing.
The same goes for the comment about the apparent contradiction in “…we investigated the performance of reversibly dynamically tuned reflective bands of ionic ferrocene chiral compounds (Figure 1, CD-Fc+) doped with negative LCs…” (lines 68-69 in the revised version) and “..we prepared a class of ferrocene derived ionic chiral compound CD-Fc+, and used it as a dopant to prepare…” ( lines 259-260 in the new version). My comment refers to which component is the dopant? The ferrocene derivative or the liquid crystal?
As a minor detail, in the elemental analysis data (supporting information), the coefficients in the chemical formulas should be written as subscrits.
Author Response
Dear reviewer,
Thank you very much for the comments concerning the manuscript (Manuscript ID:materials-2048229). Those comments are careful, valuable and very helpful. We have read through comments carefully and have made corrections accordingly. The responses to the reviewer's comments are listed one by one and presented following:
1 I would like to thank the authors for addressing my initial comments. However, the point related to the optical rotation has not been reviewed. Although the authors state in their response that "The optical rotation of the mentioned chiral compound binaphthol has been added accordingly to the synthesis route and support materials of the revised manuscript", the optical rotation values are still missing.
Response:
Thank you very much for the careful reviewing manuscript. The optical rotation of the mentioned chiral binaphthol has been added accordingly in the “2 Materials and Methods” of the revised version.
2 The same goes for the comment about the apparent contradiction in “…we investigated the performance of reversibly dynamically tuned reflective bands of ionic ferrocene chiral compounds (Figure 1, CD-Fc+) doped with negative LCs…” (lines 68-69 in the revised version) and “..we prepared a class of ferrocene derived ionic chiral compound CD-Fc+, and used it as a dopant to prepare…” ( lines 259-260 in the new version). My comment refers to which component is the dopant? The ferrocene derivative or the liquid crystal?
Response:
Thank you very much for the careful reviewing manuscript. The mentioned inconsistent content was rewritten in the revised version. The ionic ferrocene derivatives were prepared and used as the liquid crystal dopants.
- As a minor detail, in the elemental analysis data (supporting information), the coefficients in the chemical formulas should be written as subscrits.
Response:
Thank you very much for the careful reviewing manuscript. The nonstandard coefficients in the chemical formula in the element analysis data (supporting information) has been corrected in the revised version.
If there are any other necessary for us to do, please do not hesitate to contact with me.
Best wishes.
Yours sincerely,
Wan-Li He